# OpenReview forum: "The Trojan Knowledge: Bypassing Commercial LLM Guardrails via Harmless Prompt Weaving and Adaptive Tree Search"
_ICML.cc/2026/Conference — ICML 2026 regular_

### Official Review · Reviewer_L7e2 · 2026-02-15

**Soundness:** 4
**Presentation:** 3
**Significance:** 4
**Originality:** 4
**Overall Recommendation:** 5
**Confidence:** 4

**Summary:**

This paper describes a novel jailbreak method called CKA-Agent that targets the LLMs safety guardrails by asking numerous harmless questions and stitching the answers in groups to obtain forbidden information, depicting how models are not able to detect malicious intents across multiple conversation turns. However, the paper is too focused on the attack as a diagnostic tool and not on the defense solution, leaving the context-aware safeguards as a future work.

**Compliance With Llm Reviewing Policy:**

Affirmed.

**Final Justification:**

Although the information required was useful, no changes were done.

**Key Questions For Authors:**

1. How is the performance affected when using considerably smaller/weaker attack models?

2. Were humans involved in the evaluations (e.g., mainly for partial or full success type) to obtain the score?

3. What is the approx. time taken for the successful attacks vs. more simple baselines (e.g., multi-agent jailbreak)?

4. Based on your findings, it is not completely clear what specific semantic signals should future context-aware defenses monitor, please justify.

**Limitations:**

Overall, this work correctly addressed the limitations. However, the paper is too focused on the attack as a diagnostic tool and not on the defense solution, leaving the context-aware safeguards as a future work, and this should be probably mentioned as well.

**Strengths And Weaknesses:**

- Soundness: this work is technically rigorous due to having a strong theoretical core, compelling claims and well-developed experiments, including an appropriate methods and justified conclusions.

- Presentation: the presentation follows the logical structure although the figures fill a bit too small and packed. On the other hand, detailed appendices facilitate reproducibility.

- Significance: it addresses a critically important vulnerability that may reshape the current red-teaming research direction towards an emerging path.

- Originality: compared to existing research, it feels novel due to being distinct from decomposition-based attacks.

---

> ### Author Rebuttal · Authors · 2026-03-30
>
> We sincerely thank Reviewer L7e2 for the thoughtful and supportive review, for recognizing the technical rigor, significance, and originality of our work. Below, we provide detailed responses to each point.
>
> > Q1. Performance under smaller / weaker attack models.
>
> To examine this point, we replaced the attack agent in the CKA-Agent framework, which is responsible for the adaptive decomposition and exploration, from huihui-ai/Qwen3-32B-abliterated to the smaller huihui-ai/Qwen3-8B-abliterated model. Under the setting of StrongREJECT, with GPT-oss-120B as the target model and Gemini-2.5-Flash as the judge, CKA-Agent remains highly effective: the full success rate changes from 0.982 to 0.969, with partial success = 0.019, vacuous = 0.012, and refusal = 0.000. These results suggest that as long as the attack agent retains reasonably good reasoning ability, even a weaker attacker with reduced prior knowledge can still leverage the target model’s knowledge to perform effective adaptive search and achieve a very high jailbreak success rate. We will include this ablation in the revised version to better characterize attacker-strength sensitivity.
>
> > Q2. Whether human evaluation was involved.
>
> Yes, we included human evaluation to validate our scoring pipeline. Our main results use LLM-as-a-Judge for scalability, but we examined its rigor from two angles. First, we compared results across multiple strong judges (Gemini-2.5-Pro, GPT-5-mini, and benchmark-provided fine-tuned judges for HarmBench and StrongREJECT). As reported in Table 4, results are highly consistent across judges, with CKA-Agent consistently achieving the strongest performance. Second, we conducted a human evaluation study (Figure 7) comparing two settings to examine how well automatic judgments align with human assessment. The results provide supporting evidence that the judge-based FS/PS/V/R labels are reasonably well aligned with informed human evaluation. Overall, our evaluation is supported by both cross-judge consistency and human validation, strengthening the reliability of reported scores.
>
> > Q3. What is the approximate time taken for the successful attacks vs. more simple baselines?
>
> We thank the reviewer for this important question. We measured the average wall-clock time per successful sample. In the representative setting of HarmBench on Gemini-2.5-Pro, CKA-Agent takes about 740.1s per successful sample, compared with 275.8s for the Multi-Agent. At the same time, CKA-Agent is faster than several iterative / multi-turn baselines such as X-Teaming, PAIR, AutoDAN, and Actor-Attack. We summarize the results below.
>
> | Method        | Successes | Avg time / successful sample (s) |
> |-|-|-:|
> | Multi-Agent   | 103/126   | 275.8 |
> | CKA-Agent     | 122/126   | 740.1 |
> | X-Teaming     | 96/126    | 1243.5 |
> | Actor-Attack  | 41/126    | 824.4 |
> | PAIR          | 114/126   | 829.2 |
> | AutoDAN       | 116/126   | 881.3 |
>
> Overall, CKA-Agent‘s time cost remains moderate among adaptive attack methods while achieving significantly stronger attack performance.
>
> > Q4 / L1. It is not completely clear what specific semantic signals should future context-aware defenses monitor, please justify. Clarifying the scope and limitation regarding defense.
>
> We agree that defending against malicious intent distributed across multiple benign-looking knowledge fragments is an important challenge raised by our findings. We also agree that the current paper focuses more on diagnosing this failure mode than on developing a complete defense, and we will clarify this scope more explicitly in the revision.
>
> Our results suggest that future context-aware defenses should monitor not only individual prompt risk, but also whether multi-turn knowledge fragments jointly become sufficient to form a harmful information closure. In CKA-Agent, no single query explicitly contains the harmful objective; rather, the risk emerges when individually innocuous fragments accumulate and become sufficient for downstream synthesis. Thus, the key semantic signal is cross-turn compositionality, rather than isolated prompt toxicity.
>
> This further suggests a concrete defense principle: the monitor should operate at the turn-sequence level and intervene once the accumulated context exposes enough complementary information to enable harmful synthesis. From a training perspective, CKA-Agent can also serve as a useful data generator for future safeguards, since its tree-structured decomposition naturally produces diverse multi-turn trajectories with distributed malicious intent. The defense label can then be assigned to the earliest turn at which the collected fragments become sufficient for synthesis.
>
> Overall, we believe our findings provide a concrete direction for future safeguards: monitor cross-turn semantic composition and detect when benign fragments collectively become dangerous. We will revise the paper to make this implication and the scope limitation more explicit.

---

> > ### Author Rebuttal · Reviewer_L7e2 · 2026-04-02
> >
> > Thanks for your additional justification and tests, now I don't have more questions. Best.

---

> > > ### Author Response · Authors · 2026-04-04
> > >
> > > We deeply appreciate Reviewer L7e2 for the generous support and valuable feedback throughout the review process. We will include the newly added timing analysis in the final version to make the paper more complete.

---

### Official Review · Reviewer_M1v2 · 2026-03-08

**Soundness:** 3
**Presentation:** 3
**Significance:** 3
**Originality:** 3
**Overall Recommendation:** 5
**Confidence:** 3

**Summary:**

This paper presents the Correlated Knowledge Attack Agent (CKA-Agent), a framework that bypasses LLM safety guardrails by decomposing harmful goals into sequences of locally innocuous sub-queries. Rather than optimizing a single malicious prompt, it treats jailbreaking as an adaptive tree search over the target model’s internal knowledge base. By "weaving" together benign information fragments retrieved from the target, the agent can reconstruct prohibited knowledge, such as instructions for making explosives, without triggering intent-based detectors.

**Compliance With Llm Reviewing Policy:**

Affirmed.

**Final Justification:**

Given the authors' thorough rebuttal, most of my concerns have been addressed, and I have raised my score.

- Bias Mitigation: The clarification that the internal Node Critic (Qwen-32B) differs from the final Judge (Gemini) successfully addresses the concern regarding self-referential bias.
- Efficiency: The new system-wide token analysis confirms that the CKA-Agent is more cost-effective than prior baselines.
- Clarity & Defense: I appreciate the authors' commitment to reframing the human evaluation study and expanding the discussion on possible defenses.

The paper presents a novel paradigm for red-teaming. Given the clarified methodology and proven efficiency, I believe it makes a  contribution to the field.

**Key Questions For Authors:**

**1. Clarification of the "Self-Response" Baseline (Table 3):**
  - Could the authors provide a detailed description of the experimental setup for the Self-Response baseline? Specifically, does this involve the attack model acting as its own target by answering its own decomposed sub-queries?

**2. Internal Evaluator vs. Final Judge Independence:**
  - The CKA-Agent uses Gemini-2.5-Flash as its internal Node Critic to guide the tree search. Given that the final Judge is also the same model family, would the attack success rates remain robust if the internal evaluator and the final judge belonged to entirely different model families to avoid self-referential bias?

**3. Total Computational Transparency (Appendix D.2):**
  - Do the efficiency metrics reported in Appendix D.2 include the token and API consumption of the internal Decomposer, Evaluator, and Synthesizer modules? Providing a total system-wide cost (orchestration + target interaction) would ensure a fairer comparison with other agentic methods like TAP.

**Limitations:**

Yes

**Strengths And Weaknesses:**

### Strengths
- **Novel and Intuitive Paradigm:**
  - The shift from prompt optimization to internal knowledge correlation is a highly logical and novel approach to bypassing safety guardrails. It effectively exploits the fact that while models are trained to refuse harmful intent, they struggle to aggregate intent distributed across individually harmless sub-tasks.
- **Comprehensive Experimental Validation:**
  - The authors provide a rigorous evaluation across four state-of-the-art commercial LLMs (Gemini-2.5-Flash/Pro, GPT-oss-120B, and Claude-Haiku-4.5) using high-stakes benchmarks like HarmBench and StrongREJECT.

### Weaknesses
- **Self-Referential Bias in Attack and Evaluation Models:**
  - The CKA-Agent uses Gemini-2.5-Flash as its internal Node Critic to guide the search, which is the same model family used for the final Judge. This setup creates a potential self-referential bias, as the attack is optimized based on the specific safety thresholds of the model used to evaluate it. This also makes for an uneven comparison with static baselines like Multi-Agent Jailbreak, which lack a real-time judge to steer their search toward success.
- **Incomplete Reporting of Total Computational Costs (Appendix D.2):**
  - While the paper reports efficiency in terms of interactions with the Target LLM, it does not clearly account for the total token consumption of the Agent framework itself. The CKA-Agent involves multiple internal LLM calls for decomposition, evaluation, and path selection at every node, often processing long conversation histories. A transparent report of the total system-wide costs would allow for a fairer comparison with other agentic methods like TAP and Multi-Agent Jailbreak.
- **Methodological Concern Regarding Human Calibration (Appendix D.5):**
  - The authors' claim that the judge model is "well-calibrated and aligns closely with expert human judgment" is methodologically problematic. By providing the judge’s internal reasoning to evaluators who lacked specific expertise in chemistry or biology, the study introduces a significant anchoring bias. Instead of providing an independent gold-standard validation, the jump in Spearman correlation from 0.52 to 0.90 suggests that the human evaluators were persuaded to fit the model's pre-existing conclusions. Consequently, the results demonstrate the persuasiveness of the model's logic rather than its objective accuracy or calibration.
- **Opportunity for Defense Discussion:**
  - The paper identifies a significant vulnerability where intent is distributed across innocuous sub-queries. While the focus is on red-teaming, the manuscript would benefit from a brief discussion or prototype of a proactive defense. Exploring context-aware guardrails that aggregate intent across a full conversation trajectory would provide valuable actionable insights for the defensive community.

---

> ### Author Rebuttal · Authors · 2026-03-30
>
> We sincerely thank Reviewer M1v2 for the thoughtful review and for recognizing the novelty of our paradigm and the breadth of our experiments. Below, we respond to each point.
>
> > W1 / Q2. Potential self-referential bias between the internal node evaluator and the final judge.
>
> We would like to clarify that the internal Node Critic is not the same model as the final judge. As stated in Lines 214–216, the Node Critic is instantiated with the same model as the attacker (e.g., Qwen-32B), not Gemini-2.5. The final judge is separate, e.g., Gemini-2.5-Flash/Pro, GPT-5-mini, or the benchmark-provided judges in Table 4. Thus, the attack is not directly optimized against the model family used for final evaluation.
>
> > W2 / Q3. Clarifying total system-wide computational cost beyond target-model interactions.
>
> We agree that full system-level accounting is useful for fair comparison with other agentic methods. Our original cost analysis in Appendix D.2 focused on interactions with the closed-source target model and online judge, since these are the main API-billed costs in realistic black-box settings. By contrast, the attacker side uses a self-hosted open-source model.
>
> To address this, we additionally analyze attacker-side token usage. On HarmBench with Gemini-2.5-Pro, CKA-Agent uses ~4.6k attacker tokens per example on average. For comparison: Multi-Agent ~2.1k, Actor-Attack ~12.3k, AutoDAN ~12.3k, PAIR ~26.1k, X-Teaming ~133.3k. Thus, CKA-Agent's full system-wide cost remains moderate and substantially lower than several iterative baselines. We will add this analysis in the revision.
>
> > W3. Methodological concern regarding human calibration.
>
> We thank the reviewer for this thoughtful comment. We agree that this is an important methodological concern, and we do not intend to over-claim the human study as an independent gold-standard validation of objective correctness.
>
> Jailbreak evaluation is inherently subjective. For this reason, the primary evidence for evaluation robustness in our paper comes from cross-judge consistency across multiple strong judges, including both frontier proprietary judges and benchmark-provided judges, which show a highly consistent overall trend.
>
> Our human evaluation in Fig. 7 serves as a complementary analysis that examines the alignment between human judgment and automatic scoring. We believe both human evaluation and LLM-based judging are important and offer different strengths: human evaluation provides grounded, interpretable assessments, while LLM judges enable scalable and reproducible evaluation. The experiment in Fig. 7 shows that providing additional reasoning/context improves the agreement between human evaluators and the automatic judge, suggesting that the two approaches are well aligned when evaluators have access to sufficient information. We will revise the wording in the paper to better reflect this complementary relationship.
>
> > W4. Opportunity for defense discussion.
>
> We agree the manuscript would benefit from clearer defense discussion. Our main focus is diagnosing a previously underexplored attack-side failure mode; the defense direction is briefly noted in Appendix G and will be made more explicit.
>
> Our key finding is that harmful intent can be distributed across innocuous sub-queries and emerge only after cross-turn aggregation. This suggests future context-aware safeguards should monitor whether multi-turn fragments jointly become sufficient for harmful synthesis, the key signal is cross-turn semantic composition, not prompt-level toxicity alone.
>
> CKA-Agent can also serve as a training data generator for such defenses, as its tree-structured decomposition produces diverse multi-turn trajectories with distributed intent. A defense label can be assigned at the earliest turn where fragments become sufficient for synthesis.
>
> > Q1. Clarification of the Self-Response baseline.
>
> We thank the reviewer for this question. In our paper, Self-Response means directly prompting the attacker model itself to answer the harmful request, and then evaluating that response with the same judge, i.e., without interacting with the closed-source target. To further clarify this setup, we also tested a variant that keeps the same CKA-style decomposition framework but replaces the closed-source target with the attacker itself. On HarmBench, this variant gives results very close to Self-Response: about +5% under Gemini-Flash evaluation and about -2% under the stricter Gemini-Pro evaluation. This suggests that once the closed-source target is removed, the CKA scaffold alone does not materially change performance; the main gain comes from eliciting and weaving knowledge revealed by the target model.

---

> > ### Author Rebuttal · Reviewer_M1v2 · 2026-04-02
> >
> > My concerns have been adequately addressed. I have raised my score accordingly.

---

> > > ### Author Response · Authors · 2026-04-04
> > >
> > > We sincerely thank Reviewer M1v2 for the valuable feedback and for kindly raising the score. We will further improve the paper based on the reviewer's suggestions in the final version.

---

### Official Review · Reviewer_8k2q · 2026-03-08

**Soundness:** 2
**Presentation:** 3
**Significance:** 3
**Originality:** 2
**Overall Recommendation:** 4
**Confidence:** 4

**Summary:**

This work proposes jailbreaks that decompose a harmful objective into a sequence of locally harmless queries and argues that current guardrails often fail to aggregate intent across such interactions. It designs CKA-Agent, which formulates jailbreaking as adaptive exploration of a target model's correlated knowledge via UCT-guided tree search, hybrid node evaluation, and final synthesis over retrieved fragments. Across selected high-stakes subsets of HarmBench and StrongREJECT, as well as four commercial target models, the method substantially outperforms prompt-optimization and static decomposition baselines.

**Compliance With Llm Reviewing Policy:**

Affirmed.

**Final Justification:**

My concerns have been well addressed. Please update the figures in the CR version.

**Key Questions For Authors:**

- Comparison with DAMON and some guardrail-bypassing baselines.
- Explore minimum attacker capability.
- Evaluate adaptive defenses.
- See weaknesses.

**Limitations:**

Yes.

**Strengths And Weaknesses:**

- Pros
  - The reasoning-DAG framing provides a clear conceptual lens for distinguishing prompt optimization, static decomposition, and adaptive decomposition.
  - The pipeline is well organized and generally easy to follow.
- Cons
  - Its novelty is somewhat incremental relative to recent multi-turn decomposition and tree-search jailbreak work, particularly relative to the MCTS-based jailbreak approaches (e.g., DAMON).
  - It considers guardrail bypassing as a contribution, yet omits SoTA methods specifically designed to evade guardrail detection (e.g., JAM, FlipAttack, and JAIL-CON).
  - The reasoning-DAG view is intuitive, but it remains primarily a conceptual interpretation. It does not directly validate that the explored search structure corresponds to internal knowledge organization in the target models.
  - The defense evaluation is insufficient. Since the paper's core claim concerns cross-turn intent aggregation, stronger context-aware or session-level defenses would be the most relevant missing comparison.
  - There is not enough discussion of where CKA-Agent fails, how robust it is to hyperparameter choices, etc.
  - The evaluation is limited to partial prompts across selected high-stakes categories. This narrow coverage may inflate success rates compared to a broader benchmark and limit the generalizability of claims. The authors should evaluate on the full HarmBench and StrongREJECT benchmarks.
  - The attack model universally takes Qwen3-32B-abliterated (a model with its safety training surgically removed), which represents an unrealistic yet highly capable attacker. The minimum attacker capability required for the framework to function is acknowledged as an open question but not empirically characterized, making it difficult to assess real-world threat magnitude.
  - The empirical claims still rely heavily on LLM-as-a-Judge evaluation. Although it includes judge-consistency analysis and a small human study, the human validation remains limited in scale and depends strongly on exposing the judge's reasoning.
  - The LLM it evaluates is slightly outdated and lacks some frontier models such as GPT-5.X and Gemini-3.
  - The font size in Figures 1-3 is too small.

---

> ### Author Rebuttal · Authors · 2026-03-30
>
> We sincerely thank Reviewer 8k2q for the constructive feedback. Due to length limits, we summarize below and provide full tables at https://anonymous.4open.science/r/CKA-Agent-ICML/Rebuttal.md
>
> > W1 / Q1. Distinction from prior tree-search jailbreaks (eg. DAMON).
>
> The key difference is that prior tree-search attacks search over attacker-side prompts or static decomposition strategies, while CKA-Agent searches over correlated knowledge fragments dynamically revealed by the target model. TAP performs tree-based search over attacker-generated prompt refinements; DAMON introduces MCTS over dialogue states, but its search space remains over attacker-designed static decomposition strategies. In contrast, CKA-Agent treats the target as a knowledge oracle, issues locally innocuous queries, and reconstructs the harmful answer only after sufficient fragments are dynamically collected.
>
> Empirically, under GPT-oss-120B on StrongREJECT, DAMON achieves FS=0.654 while CKA-Agent achieves FS=0.982 (Table R1), supporting that the gain comes from target-revealed knowledge.
>
> > W2. Add guardrail-evasion baselines (JAM, FlipAttack, JAIL-CON).
>
> We evaluated all three suggested baselines under StrongREJECT with GPT-oss-120B. The strongest among them, JAIL-CON, achieves only FS=0.278, while CKA-Agent achieves FS=0.982 (Table R2). These methods are weaker than many recent baselines in our paper under today's stronger guardrails. This further confirms that weaving distributed benign-looking knowledge fragments is more effective than prior prompt-transformation or guardrail-evasion techniques.
>
> > W3. Clarifying the role of the reasoning-DAG interpretation.
>
> Our reasoning-DAG view is a structured lens for characterizing how correlated knowledge fragments are adaptively decomposed, explored, and synthesized during jailbreak search. It provides a principled way to model the search space induced by distributed malicious intent. Prior work supports this perspective: factual knowledge in LLMs is mediated by structured internal dependencies rather than flat isolated facts, and graph-structured reasoning signals are measurably present in model internals [1,2]. We note that our claim is not that we have fully recovered the target model's true internal ontology, this is a limitation we acknowledge. Rather, the reasoning-DAG abstraction is both interpretable and empirically useful: it guides an effective adaptive attack and helps explain why distributed knowledge-fragment weaving can bypass current safeguards.
>
> [1] The Evolution of Latent Representations in a Dynamic Knowledge Graph
>
> [2] Probing the Graph Structure of Reasoning in LLMs
>
> > W4 / Q3. Context-aware defense evaluation.
>
> Our paper includes CKA-Agent-Branch (Table 3), where the model receives full branch history. We further implemented CKA-Agent-Primed based on Intention Analysis [3]: the target explicitly analyzes whether sub-queries may jointly encode malicious intent before responding. Even with this stronger defense, Full Success remains 0.883/0.852/0.821/0.864 on Gemini-Flash/Pro/GPT-oss/Claude-4.5 on StrongREJECT (Table R3), suggesting current models still lack robust cross-turn aggregation against distributed malicious intent.
>
> [3] Intention Analysis Makes LLMs A Good Jailbreak Defender
>
> > W5 / W6. Hyperparameter robustness, failures, and full-benchmark evaluation.
>
> Full tables (R4-R6) are in our link.
>
> **Hyperparameter robustness.** Except for results in our Appendix, we further ablated maximum search depth on StrongREJECT (GPT-oss-120B). FS improves from 0.939 (depth=3) to 0.982 (depth=5) then saturates at 0.988 (depth=7), indicating CKA-Agent is not overly sensitive to this choice.
>
> **Failure analysis.** The few remaining failures concentrate in norm-sensitive categories rather than concrete knowledge-grounded ones.
>
> **Full benchmark coverage.** We ran CKA-Agent on the complete StrongREJECT dataset and achieved FS=0.952, consistent with subset results. Our original subset selection prioritizes categories that better align with our core motivation of complex correlated knowledge decomposition.
>
> > W7 / Q2. Characterizing attacker-strength sensitivity.
>
> We evaluated CKA-Agent with a weaker attacker (Qwen3-8B replacing 32B); FS only drops from 0.982 to 0.969. See our response to Reviewer L7e2 Q1 for details.
>
> > W8. The empirical claims still rely heavily on LLM-as-a-Judge evaluation.
>
> Our claims do not rely on a single judge. Table 4 reports consistent results across Gemini-2.5-Flash/Pro, GPT-5-mini, and benchmark-provided fine-tuned judges, with stable method rankings. We also include human evaluation as complementary validation. See our response to Reviewer M1v2 W3 for details.
>
> > W9. Extending evaluation to newer frontier models.
>
> CKA-Agent on GPT-5.2 and Gemini-3.0-Pro achieves FS=0.932/0.951 (StrongREJECT) and 0.889/0.881 (HarmBench) (See Table R7), confirming generalization to newer models.
>
> > W10. Improving figure readability.
>
> We will enlarge text in the revision.

---

> > ### Author Rebuttal · Reviewer_8k2q · 2026-04-02
> >
> > Thank you for the reply. However,
> >
> > 1. The current answer still doesn't clarify the specific technical differences between this method and DAMON. The technical novelty explained by the authors isn't significant compared to previous work. I suggest adding specific visual comparison images or tables, which might be helpful.
> >
> > 2. Your main defense experiment (Figure 4) is based on Gemini-2.5-Flash. Why was GPT-OSS-120B chosen when comparing it to guardrail-evasion baselines? Furthermore, their performance against defenses should be included because they are designed to bypass defenses.
> >
> > Minors: Please provide the revised figures via an anonymous link.

---

> > > ### Author Response · Authors · 2026-04-04
> > >
> > > We thank Reviewer 8k2q for the thoughtful follow-up questions. Below we address each point with additional clarification, comparisons, and supporting results. All follow-up experiments, and tables are available at the anonymous link: https://anonymous.4open.science/r/CKA-Agent-ICML/followup_response.md
> > >
> > > > Q1. Further detailed comparison between CKA-Agent and DAMON.
> > >
> > > We thank the reviewer for this helpful question. To make the distinction more explicit, we provide a more detailed side-by-side comparison between CKA-Agent and DAMON in Table R1 of the anonymous follow-up link, including their core methodological differences, and real examples. We also provide a same-setting empirical comparison on the strong guardrail model GPT-5.2 (In Table 1). Under the same StrongREJECT / GPT-5.2 setting, DAMON achieves only 0.275 full success, whereas CKA-Agent reaches 0.932, showing that the gain is not merely due to generic tree search, but reflects a more fundamental difference between prior strategy-based search and our adaptive knowledge-weaving framework.
> > >
> > > **Table 1.** Same-setting comparison between DAMON and CKA-Agent on StrongREJECT with GPT-5.2 as the target model.
> > >
> > > | Method    | FS↑      | PS↑    | V↓     | R↓     |
> > > |-----------|----------|--------|--------|--------|
> > > | DAMON     | 0.2754   | 0.1856 | 0.0180 | 0.5210 |
> > > | CKA-Agent | **0.9320** | 0.0560 | 0.0060 | 0.0060 |
> > >
> > > > Q2 (a). Clarifying the use of different target models in the baseline comparison and defense evaluation settings.
> > >
> > > We thank the reviewer for this helpful question. The two settings are designed to answer different empirical questions.
> > >
> > > For the comparison with guardrail-evasion baselines in Table R2 (https://anonymous.4open.science/r/CKA-Agent-ICML/Rebuttal.md), our goal is to test whether the advantage of CKA-Agent remains substantial under a strong native guardrail system. In our paper’s main evaluation, we compare multiple target models, including Gemini-2.5-Flash, Gemini-2.5-Pro, GPT-oss-120B, and Claude-Haiku-4.5, and observe that GPT-oss-120B and Claude-Haiku-4.5 are substantially harder targets, showing stronger refusal/safety behavior against prior attacks. Table R2 further shows that, even in this stronger-guardrail setting, CKA-Agent still substantially outperforms prior guardrail-evasion baselines such as JAM, JAIL-CON, and FlipAttack. This choice is also broadly consistent with some public safety evaluations; for example, EnkryptAI’s safety leaderboard.
> > >
> > > By contrast, Figure 4 (original paper) studies a different question: the effectiveness of additional defense strategies augmented on top of a target model, rather than the base model’s native guardrail strength itself. In this setting, Gemini-2.5-Flash serves as a clearer testbed for isolating the marginal effect of the added defenses. If the base model already has an extremely strong built-in guardrail, the incremental effect of the augmented defense can be harder to interpret cleanly. Therefore, the use of different target models reflects different experimental goals, rather than an inconsistency in evaluation.
> > >
> > > > Q2 (b). Evaluating guardrail-evasion baselines under the same defense settings as Figure 4.
> > >
> > > We thank the reviewer for this helpful suggestion. We further study this question in the Figure 4 (Original Paper) defense setting by evaluating the reviewer-suggested baselines, JAM, JAIL-CON, FlipAttack, together with CKA-Agent, under the same augmented defenses on Gemini-2.5-Flash. In our paper, Circuit Breakers and LLM Guard are the two most effective defense strategies; therefore, we use these two defenses here to provide the most relevant comparison under an identical protocol. We report the detailed results in Table R2 of the anonymous follow-up response link.
> > >
> > > The results show that prior guardrail-evasion baselines remain substantially weaker than CKA-Agent under both defenses. Under Circuit Breakers, FlipAttack and JAIL-CON are almost fully suppressed, with full success rates of 0.000 and 0.000, respectively, while JAM reaches only 0.219. In contrast, CKA-Agent still achieves a full success rate of 0.873. Under LLM Guard, FlipAttack, JAIL-CON, and JAM achieve only 0.016, 0.071, and 0.230 full success, respectively, whereas CKA-Agent reaches 0.984 full success.

---

### Official Review · Reviewer_edQ4 · 2026-03-12

**Soundness:** 2
**Presentation:** 3
**Significance:** 2
**Originality:** 2
**Overall Recommendation:** 4
**Confidence:** 3

**Summary:**

This paper studies how safety-aligned large language models can still reveal restricted or harmful knowledge through structured prompting strategies. The authors argue that alignment methods suppress unsafe outputs but do not eliminate internal representations of harmful knowledge. The main contribution is an attack methodology called Correlated Knowledge Attack Agent (CKAAgent) that bypasses surface-level safety protections by decomposing or reframing harmful queries. The authors show that aligned models can disclose restricted content when prompted indirectly, even when direct prompts are refused. The results show attack success across multiple aligned models, suggesting that safety alignment does not remove internal harmful knowledge.

**Compliance With Llm Reviewing Policy:**

Affirmed.

**Final Justification:**

The rebuttal addresses my concerns because the authors clarify the conceptual distinction of their approach and provide additional targeted ablations that strengthen the empirical understanding of the method. While some minor limitations remain in baseline coverage, they do not affect the core contribution. Overall, the rebuttal increases my confidence in the paper, and I raise my score accordingly.

**Key Questions For Authors:**

Questions are mentioned in the weaknesses.

**Limitations:**

Yes.

**Strengths And Weaknesses:**

Strengths
- The paper specifically highlights a safety issue: alignment may suppress outputs without modifying the underlying knowledge distribution. The authors evaluate multiple models and baselines, suggesting the phenomenon is not model-specific. The results further support the claim that alignment training may not remove dangerous capabilities.
- The paper is clearly written and well structured, which makes the reader easy to follow.

Weaknesses
- In terms of originality, although the paper claims to introduce a structured search-based attack framework, the distinction between this method and prior reasoning-based jailbreak techniques is not clearly articulated. For example, many previous works also rely on exploring reasonings or gradually extracting restricted knowledge, which weakens the originality claim. [1,2]
- The framework includes only a limited ablation comparing the full system with a standalone agent. It does not provide component-level ablations to analyze the contribution of key design choices such as tree search depth or decomposition modules.

[1]: Reasoning-Augmented Conversation for Multi-Turn Jailbreak Attacks on Large Language Models

[2]: Great, Now Write an Article About That: The Crescendo Multi-Turn LLM Jailbreak Attack

---

> ### Author Rebuttal · Authors · 2026-03-30
>
> We sincerely thank Reviewer edQ4 for the constructive feedback, as well as for recognizing the significance of our safety findings. Below, we provide detailed responses to the questions raised.
>
> > W1. Key differences between CKA-Agent and prior reasoning-based jailbreak methods.
>
> We thank the reviewer for this comment. Our novelty claim is not simply “multi-turn” or “decomposition.” Rather, most prior multi-turn jailbreaks, including RACE [1], Crescendo [2] and its follow-up ActorBreaker, as well as the recent DAMON [3], are fundamentally strategy-based: they rely on attacker-designed strategies such as role-play, purpose inversion, or attacker-planned static decomposition, and when blocked, they switch to another reformulation. In contrast, CKA-Agent formulates jailbreak as a reasoning-DAG search problem over correlated knowledge fragments. The target model is treated as a knowledge oracle; each target-facing query is only a locally innocuous fragment, and the harmful answer is reconstructed only after sufficient fragments have been dynamically collected.
>
> This distinction is also reflected empirically. RACE reports HarmBench scores of 62.5 on Gemini 2.0 Flash and 82.8 on GPT-4o in its Table 1, while our method achieves 96–97% full success on substantially stronger guardrail models such as Gemini-2.5-Pro, GPT-oss-120B, and Claude-Haiku-4.5 [1]. We also note that the authors state in the public RACE GitHub repository that the code is not yet open-sourced, so we are currently unable to perform a direct reproduction from their official implementation. Our paper already includes ActorBreaker, a recent follow-up to Crescendo [2], and it shows a clear gap from CKA-Agent on strongly aligned models. In addition, following Reviewer 8k2q’s suggestion, we newly evaluated DAMON [3] under the same StrongREJECT / GPT-oss-120B setting:
>
> | Method     | Full Success | Partial Success | Vacuous | Refusal |
> |-|-|-|-|-|
> | DAMON      | 0.654 | 0.111 | 0.068 | 0.167 |
> | CKA-Agent  | 0.982 | 0.012 | 0.006 | 0.000 |
>
> Overall, we will revise the paper to make this distinction explicit. The updated comparison still suggests that current strong-guardrail models remain significantly more vulnerable to knowledge-fragment weaving than to prior strategy-based multi-turn attacks.
>
> [1] Reasoning-Augmented Conversation for Multi-Turn Jailbreak Attacks on Large Language Models
>
> [2] Great, Now Write an Article About That: The Crescendo Multi-Turn LLM Jailbreak Attack
>
> [3] DAMON: A Dialogue-Aware MCTS Framework for Jailbreaking Large Language Models
>
> > W2. Additional ablation on individual components
>
> We thank the reviewer for this valuable suggestion. We agree that more explicit component-level ablations help clarify the contribution of CKA-Agent’s key design choices.
> We would like to note that the current paper includes several analyses beyond the full-system comparison. Specifically, Table 3 compares the attacker’s standalone self-response with the full CKA-Agent system, which helps isolate the role of target-guided search and target feedback. Table 2 further studies CKA-Agent-Branch, providing additional evidence on intent distribution under single-session history exposure. In addition, the appendix analyzes the search process itself: Fig. 5 shows clear multi-iteration gains from adaptive branching, while Fig. 6 shows that these gains are achieved with a favorable cost–performance trade-off rather than simply by increasing interaction budget.
>
> To address the reviewer’s concern more directly, we have added finer-grained ablations on two important design factors: search depth and attacker strength. First, we vary the maximum search depth while keeping the rest of the framework fixed. Under StrongREJECT / GPT-oss-120B, increasing the depth from 3 → 5 → 7 improves full success from 0.939 → 0.982 → 0.988, showing that shallow search is weaker, while the gain begins to saturate beyond our default choice of depth = 5. Second, we replace the original attacker huihui-ai/Qwen3-32B-abliterated with the smaller huihui-ai/Qwen3-8B-abliterated model. Even under this weaker attacker, CKA-Agent still achieves 0.969 full success, indicating that the framework’s gain does not depend on one particularly strong attacker instantiation.
>
> | Ablation Setting | Configuration | FS↑   | PS↑   | V↓    | R↓    |
> |-|-|-|-|-|-|
> | Depth     | Depth = 3     | 0.939 | 0.049 | 0.006 | 0.006 |
> | Depth     | Depth = 5     | 0.982 | 0.012 | 0.006 | 0.000 |
> | Depth     | Depth = 7     | 0.988 | 0.012 | 0.000 | 0.000 |
> | Attacker  | Qwen3-8B-abliterated | 0.969 | 0.019 | 0.012 | 0.000 |
>
> Overall, these results support the same conclusion: the effectiveness of CKA-Agent is not explained by a particular attacker or by longer interaction budget alone. Rather, it comes from the combination of adaptive multi-step search, target-correlated knowledge fragment-guided exploration, and distributed decomposition over correlated benign fragments.

---

> > ### Author Rebuttal · Reviewer_edQ4 · 2026-04-03
> >
> > The rebuttal addresses my concerns well. In particular, the authors clarify the distinction from prior reasoning-based methods and provide component-level ablations that strengthen the understanding of the framework. I will increase my score.

---

> > > ### Author Response · Authors · 2026-04-04
> > >
> > > We sincerely thank Reviewer edQ4 for the thoughtful engagement with our rebuttal and for kindly raising the score. We will incorporate the clarifications on the distinction from prior reasoning-based methods and the component-level ablations into the final version.

---

### Decision · Program_Chairs · 2026-04-30

**Decision:**

Accept (regular)

**Comment:**

2 x Accept (5), 2 x Weak accept (4). This paper introduces CKA-Agent, a jailbreak framework that reframes the attack as an adaptive, tree-structured exploration over correlated knowledge fragments, decomposing harmful objectives into locally innocuous sub-queries to bypass commercial LLM guardrails. During the rebuttal and discussion phase, the authors clarified the distinction from prior reasoning-based and tree-search attacks, added reviewer-suggested guardrail-evasion baselines, extended evaluation to newer frontier models, provided component-level ablations, and resolved the self-referential bias concern, thereby addressing the main concerns raised during review. CKA-Agent consistently achieves high full success rates against strong commercial guardrails and substantially outperforms prior guardrail-evasion baselines under matched settings. Overall, this work identifies a previously underexplored failure mode in which malicious intent is distributed across individually innocuous knowledge fragments, and delivers a novel and empirically compelling red-teaming paradigm that makes a timely contribution to LLM safety.